# NKB signaling in the posterodorsal medial amygdala stimulates gonadotropin release in a kisspeptin-independent manner in female mice

Chrysanthi Fergani[1,2], Silvia Leon[1,2], Stephanie L Padilla[3], Anne MJ Verstegen[2,4], Richard D Palmiter[3], Victor M Navarro[1,2]*

[1]Department of Endocrinology, Diabetes and Hypertension, Brigham and Women's Hospital, Boston, United States; [2]Harvard Medical School, Boston, United States; [3]Howard Hughes Medical Institute, University of Washington, Seattle, United States; [4]Department of Medicine, Division of Endocrinology, Beth Israel Deaconess Medical Center, Boston, United States

**Abstract** Neurokinin B (NKB) signaling is critical for reproduction in all studied species. The existing consensus is that NKB induces GnRH release via kisspeptin (*Kiss1*) stimulation in the arcuate nucleus. However, the stimulatory action of NKB is dependent on circulating estrogen ($E_2$) levels, without which, NKB inhibits luteinizing hormone (LH) release. Importantly, the evidence supporting the kisspeptin-dependent role of NKB, derives from models of persistent hypogonadal state [e.g. *Kiss1r* knock-out (KO) mice], with reduced $E_2$ levels. Here, we demonstrate that in the presence of $E_2$, NKB signaling induces LH release in a kisspeptin-independent manner through the activation of NK3R (NKB receptor) neurons in the posterodorsal medial amygdala (MePD). Importantly, we show that chemogenetic activation of MePD Kiss1 neurons induces LH release, however, the stimulatory action of NKB in this area is Kiss1 neuron-independent. These results document the existence of two independent neuronal circuitries within the MePD that regulate reproductive function in females.
**Editorial note:** This article has been through an editorial process in which the authors decide how to respond to the issues raised during peer review. The Reviewing Editor's assessment is that all the issues have been addressed (see decision letter).
DOI: https://doi.org/10.7554/eLife.40476.001

*For correspondence:
vnavarro@bwh.harvard.edu

## Introduction

Reproduction is regulated by a complex neuronal network the precise components of which are still being elucidated. Nonetheless, it is well established that kisspeptin and neurokinin B (NKB) signaling systems are indispensable parts of this network. Loss-of-function mutations in the genes encoding for kisspeptin (*Kiss1*) or its receptor (*Kiss1r*) or NKB (encoded by the *Tac2* gene) and its receptor (NK3R, encoded by the *Tacr3* gene) in humans and mice, are linked to hypogonadotropic-hypogonadism and infertility (*Seminara et al., 2003*; *Topaloglu et al., 2009*).

Kisspeptins, secreted from Kiss1 neurons in the arcuate (Kiss1[ARC]) and anteroventral periventricular/periventricular (Kiss1[AVPV/PeN]) nuclei, have been directly linked to the release of gonadotropin releasing hormone [GnRH (*Smith et al., 2006*; *Fergani and Navarro, 2016*) however, there is another population of Kiss1 neurons in the posterodorsal part of the medial amygdala (Kiss1[MePD]) whose contribution to GnRH/LH stimulation has not yet been described (*Fergani and Navarro, 2016*; *Smith et al., 2006*; *Pineda et al., 2017*). In addition to kisspeptin, Kiss1[ARC] neurons express

neurokinin B and dynorphin and these neurons are sometimes referred to as KNDy neurons (*Fergani and Navarro, 2016*). Functional studies in mice and other species suggest that the stimulatory effect of NKB/NK3R signaling lies up-stream of Kiss1/Kiss1r, which in turn, directly activates GnRH neurons and, hence, stimulates luteinizing hormone (LH) and follicle stimulating hormone (FSH) secretion into the peripheral circulation (*García-Galiano et al., 2012*; *Grachev et al., 2012*; *Navarro et al., 2015*). Specifically, NKB signaling onto kisspeptin occurs via the auto-synaptic activation of NK3R residing on Kiss1[ARC] neurons (*Fergani and Navarro, 2016*). The existence of this pathway is supported by the fact that the selective NK3R agonist, senktide, induces Fos expression in Kiss1[ARC] neurons in vivo (*Navarro et al., 2011a*) and increases their electrical activity in hypothalamic slices (*de Croft et al., 2013*; *Navarro et al., 2011b*). Subsequently, kisspeptin signaling was deemed an indispensable part of the reproductive role of NKB, as the stimulatory effect of senktide on LH secretion was shown to be absent in *Kiss1r* knock out (KO) mice (*García-Galiano et al., 2012*; *Navarro et al., 2015*), prepubertal rats treated with a *Kiss1r* antagonist (*Grachev et al., 2012*) or agonadal juvenile monkeys with a desensitized *Kiss1r* (*Ramaswamy et al., 2011*). These studies clearly indicated the importance of NKB signaling onto Kiss1[ARC] neurons for GnRH/LH secretion.

Interestingly, the effect of NK3R activation via intracerebroventricular (ICV) administration of senktide on LH release is highly dependent on the sex steroid milieu; senktide was inhibitory in the absence but stimulatory in the presence of sex steroids in mice (*Navarro et al., 2015*) and sheep (*Billings et al., 2010*). This poses a predicament due to the fact that the aforementioned studies, rendering kisspeptin signaling indispensable for NKB stimulation of LH, have all been carried out in animal models characterized by a persistent hypogonadal state (i.e., in the absence of sex steroids). Furthermore, a subset of GnRH neurons has been shown to contain NK3R in rats (*Krajewski et al., 2005*) and mice (*Navarro et al., 2015*), and a kisspeptin-independent activation of GnRH neurons, by NK3R agonists, in the median eminence (ME) has been demonstrated in vitro (*Gaskins et al., 2013*). Thus, additional NKB regulation of GnRH release at a different level, that is kisspeptin-independent action, in the presence of sex steroids, cannot be excluded.

The experimental studies described here aimed to assess whether senktide can stimulate LH release, in the presence of $E_2$, in adult mice that lack a functional kisspeptin signaling system, and if so, investigate the potential mechanisms involved. We used mice in which Cre recombinase was targeted to the *Kiss1* locus and prevented Kiss1 protein synthesis; consequently, homozygous mice (*Kiss1*[cre/cre]) are *Kiss1* KO and display severe hypogonadotropic hypogonadism (*Padilla et al., 2018*). Our findings reveal a novel kisspeptin-independent pathway of GnRH/LH release, which is activated in the presence of $E_2$ and involves NKB/NK3R signaling within the MePD. Additionally, we demonstrate, for the first time, that the chemogenetic activation of Kiss1[MePD] neurons activates the gonadotropic axis and therefore, provide evidence for the existence of two independent neuronal circuitries originating from the MePD that regulate LH release in the female mouse.

## Results

### Central (ICV) administration of senktide stimulates LH release in female Kiss1 KO mice in the presence of sex steroids

To investigate potential kisspeptin-independent stimulation of LH after central activation of NK3R signaling with senktide, we compared hypogonadal *Kiss1* KO mice (*Padilla et al., 2018*) of both sexes to gonadectomized [GNX; orchidectomy ($WT_{ORX}$) or ovariectomy ($WT_{OVX}$)] adult WT male and female mice. In the absence of sex steroids [testosterone (T) or $E_2$, in males and females, respectively)] ICV senktide administration decreased plasma LH levels by ~50% in WT males ($p < 0.0001$) and ~36% in WT females ($p = 0.0016$) with no alteration observed in *Kiss1* KO mice of either sex (*Figure 1A,B*). When circulating levels of sex steroids were restored, LH release was significantly increased in WT males and females [($WT_{OVX+E2}$, $WT_{ORX+T}$); $p = 0.007$ and $p = 0.0049$, respectively; *Figure 1C,D*]. Interestingly, the same was observed in *Kiss1* KO female mice supplemented with $E_2$ [(*Kiss1* $KO_{+E2}$); $p = 0.005$; *Figure 1D*] but not in male *Kiss1* KO mice supplemented with T (*Kiss1* $KO_{+T}$; *Figure 1C*) revealing the existence of a female-specific, kisspeptin-independent but $E_2$-dependent, NKB/NK3R signaling pathway that controls LH release.

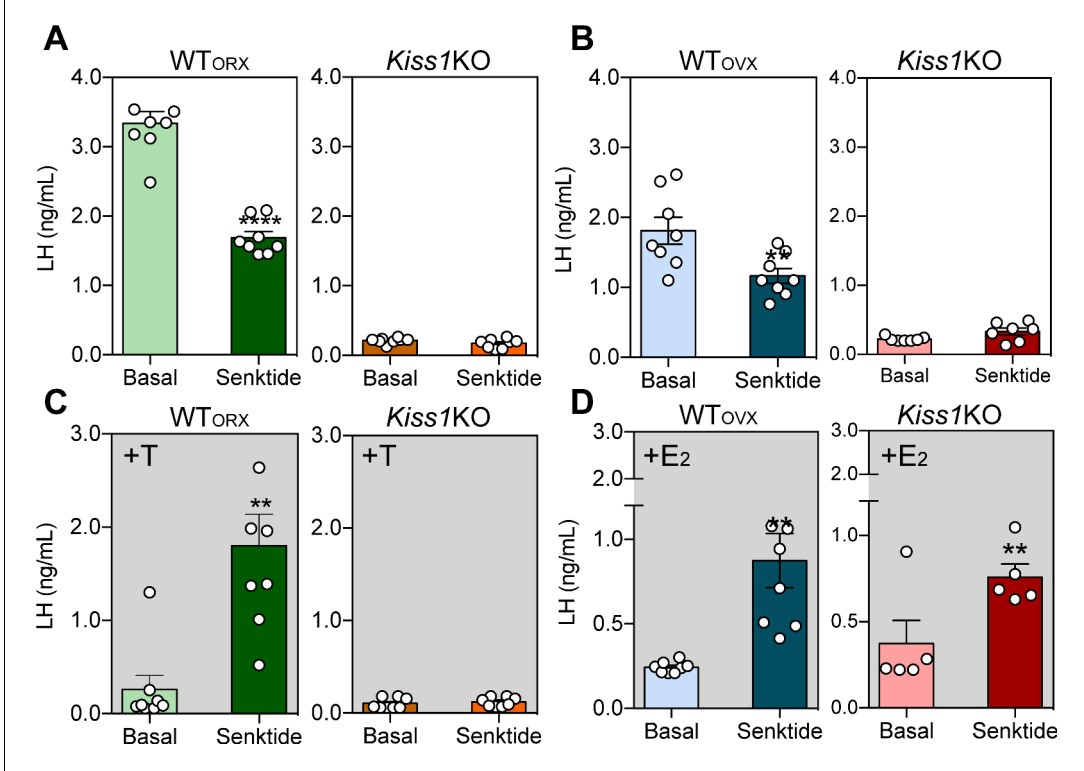

**Figure 1.** ICV injection of senktide stimulates LH release in female *Kiss1* KO mice supplemented with estradiol. Mean ± SEM LH concentrations (ng/ml) in male (A) and (C) and female (B) and (D) adult WT and homozygous $Kiss1^{Cre/Cre}$ (i.e., *Kiss1* KO) mice. Blood samples were collected before (basal) and 25 min after ICV injection of senktide (an NK3R-specific agonist; 600 pmol diluted in 5 µl 0.9% NaCl). (A, B) Mean ± SEM LH concentrations (ng/ml) in adult WT male and female mice gonadectomized ($WT_{ORX}$ or $WT_{OVX}$, respectively) and studied in parallel to hypogonadal (with low sex steroid levels) $Kiss1^{Cre/Cre}$ (i.e., *Kiss1* KO) littermates (n = 7–10/group). (C, D) Mean ± SEM LH concentrations (ng/ml) in adult gonadectomized WT and $Kiss1^{Cre/Cre}$ (i. e., *Kiss1* KO) male and female mice supplemented with testosterone or estradiol, respectively ($WT_{ORX+T}$, $WT_{OVX+E2}$, *Kiss1* $KO_{+T}$, *Kiss1* $KO_{+E2}$; n = 5–10/group). Basal *versus* after senktide injection LH concentrations were compared with a paired *t*-test ****$p < 0.0001$, **$p < 0.007$. T = testosterone, $E_2$ = estradiol.

DOI: https://doi.org/10.7554/eLife.40476.002

The following figure supplement is available for figure 1:

**Figure supplement 1.** Expression profile of (A) *Kiss1* gene in the mediobasal hypothalamus (MBH), (B) *Kiss1* gene in the preoptic area (POA), of ovariectomized (OVX) WT and hypogonadal *Kiss1 KO* female mice.

DOI: https://doi.org/10.7554/eLife.40476.003

## Chemogenetic activation of the ARC KNDy neuron stimulates LH release in control but not Kiss1 KO female mice

A Cre-dependent activating DREADD (hM3D-Gq) tagged with mCherry and packaged in an adeno-associated virus (serotype 5; *Figure 2A*) was injected into the ARC of $Kiss1^{Cre/+}$ or *Kiss1* KO (i.e. $Kiss1^{Cre/Cre}$) mice. In this animal model green fluorescent protein (GFP; together with Cre-recombinase) has been targeted to the Kiss1 locus and is, therefore, expressed solely in Kiss1 neurons. Thus, GFP immunoreactivity is identical to the detection of Kiss1 cells (*Padilla et al., 2018*) and GFP and m-Cherry colocalization represents Kiss1 cells infected with the AAV. Analysis following the completion of pharmacological studies demonstrated that mCherry expression was present throughout the ARC (*Figure 2B*) and not elsewhere. There was limited variability in the spread of mCherry among animals which extended throughout the medial-caudal extent of the ARC (approximately −1.40 mm to −2.30 mm from bregma; paxinos atlas). Six out of twenty mice (distributed among the groups) had primarily unilateral m-Cherry spread, but they did not differ significantly from their respective groups in LH concentrations, and were therefore, included in further analyses. Within the ARC, of both $Kiss1^{Cre/+}$ and *Kiss1* KO genotypes, HM3D:mCherry was expressed in ~32% of GFP-immunoreactive cells, and was not observed in non-GFP cells or other brain areas.

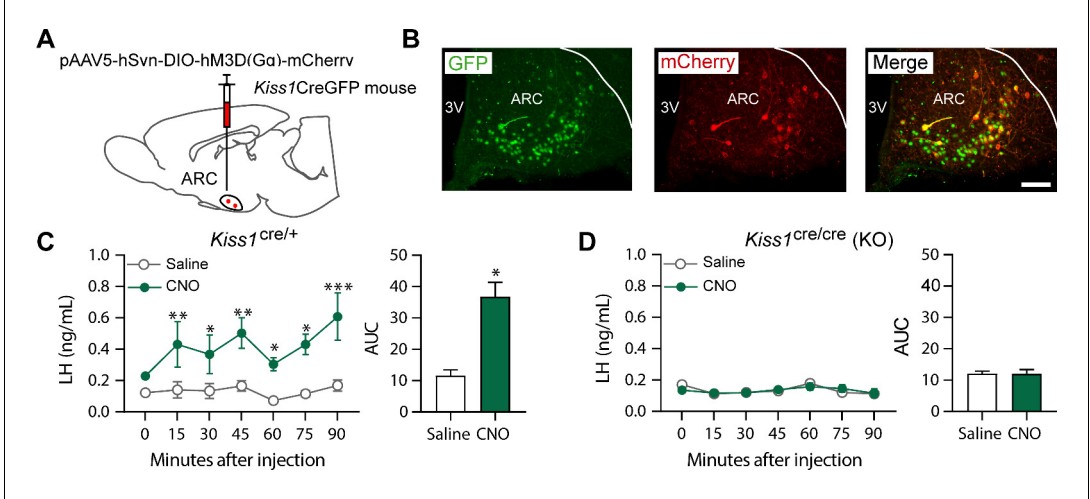

**Figure 2.** Kiss1ARC (KNDy) neuron stimulates LH release in Kiss1[Cre/+]but not Kiss1[Cre/Cre] (i.e., Kiss1 KO) mice supplemented with estradiol. (**A**) Schematic representation of the site used to inject a pAAV encoding a Cre-dependent hM3Dq DREADD tagged to mCherry (pAAV5/hSyn-DIO-hm3dq: mCherry; titer $3 \times 10^{12}$ genome copies per ml; 1 µl per hemisphere) in heterozygous Kiss1[Cre/+] (n = 10) and homozygous Kiss1[Cre/Cre] (i.e., Kiss1 KO) female mice (n = 20). (**B**) Representative photomicrograph of a coronal brain section stained for green fluorescent protein (GFP; green), mCherry (red) and merged GFP and mCherry immunoreactivity in the ARC of a homozygous Kiss1Cre/Cre (i.e., Kiss1 KO) female mouse >3 weeks after hM3Dq: mCherry injection (Scale bar 50 µm). In this animal model Cre and GFP have been targeted to the Kiss1 locus and are, therefore, expressed solely in Kiss1 neurons. Colocalization of GFP and m-Cherry immunoreactivity represents Kiss1ARC cells that have been infected with the pAAV encoding a Cre-dependent hM3Dq DREADD. (**C, D**) Mean ± SEM LH concentrations (ng/ml) and area under the curve (AUC) after an i.p. injection of saline (grey line-empty bar) or CNO (10 mg/kg dissolved in saline; green line-green bar) three weeks after the hM3Dq DREADD-injection in (**C**) heterozygous Kiss1[Cre/+] and (**D**) homozygous Kiss1[Cre/Cre] (i.e., Kiss1 KO supplemented with estradiol) female mice (n = 5–8/group). Blood samples were collected just before saline or CNO i.p. injection (0) and then every 15 min for 90 min. 3V: third ventricle, ARC: arcuate nucleus. Repeated LH concentrations at multiple time-points and between treatments were compared using a 2-WAY ANOVA and a Fishers posthoc test when appropriate. Area under the curve was compared with 2-tailed student t tests. *p < 0.035, **p < 0.006, ***p < 0.0001.

DOI: https://doi.org/10.7554/eLife.40476.004

The following figure supplement is available for figure 2:

**Figure supplement 1.** Representative merged images of mCherry (red) and GnRH (green) immunoreactivity in the ARC of *Kiss1*[Cre/+] (**A**) and *Kiss1*[Cre/Cre] (**B**).

DOI: https://doi.org/10.7554/eLife.40476.005

*Kiss1*[Cre/+] animals treated with clozapine N-oxide (CNO) showed a significant increase in LH within the first 15 min after the injection (p = 0.0057) compared to controls, which was sustained until the end of the sampling period (90 min; p < 0.0001; *Figure 2C*). However, no effect on LH release was observed in E$_2$-supplemented *Kiss1* KO (i.e. *Kiss1*[cre/cre]) animals treated with either saline or CNO (*Figure 2D*). Therefore, only mice with an intact *Kiss1* signaling system showed an increase in LH release, despite similar activation of the Kiss1[ARC] neuron in both mouse models after CNO treatment. Thus, the stimulation of LH release after senktide in *Kiss1* KO animals must occur via a different NK3R-expressing neuronal population.

## Senktide administration into the MePD, but not the ARC or POA, stimulates LH release in female WT and Kiss1 KO mice, in the presence of estrogen.

In this set of experiments, we aimed to locate the brain area which senktide may be acting to stimulate LH release. To this end, we first sought to investigate the distribution of NK3R in the WT and *Kiss1* KO mouse brain. In addition, we aimed to describe any potential anatomical interplay between NK3R and GnRH neurons, as the direct action of NKB onto the GnRH neuron is a likely kisspeptin-independent mechanism (*Figure 3*, *Figure 3—figure supplement 2*, *Figure 4* and *Figure 4—figure supplement 1*).

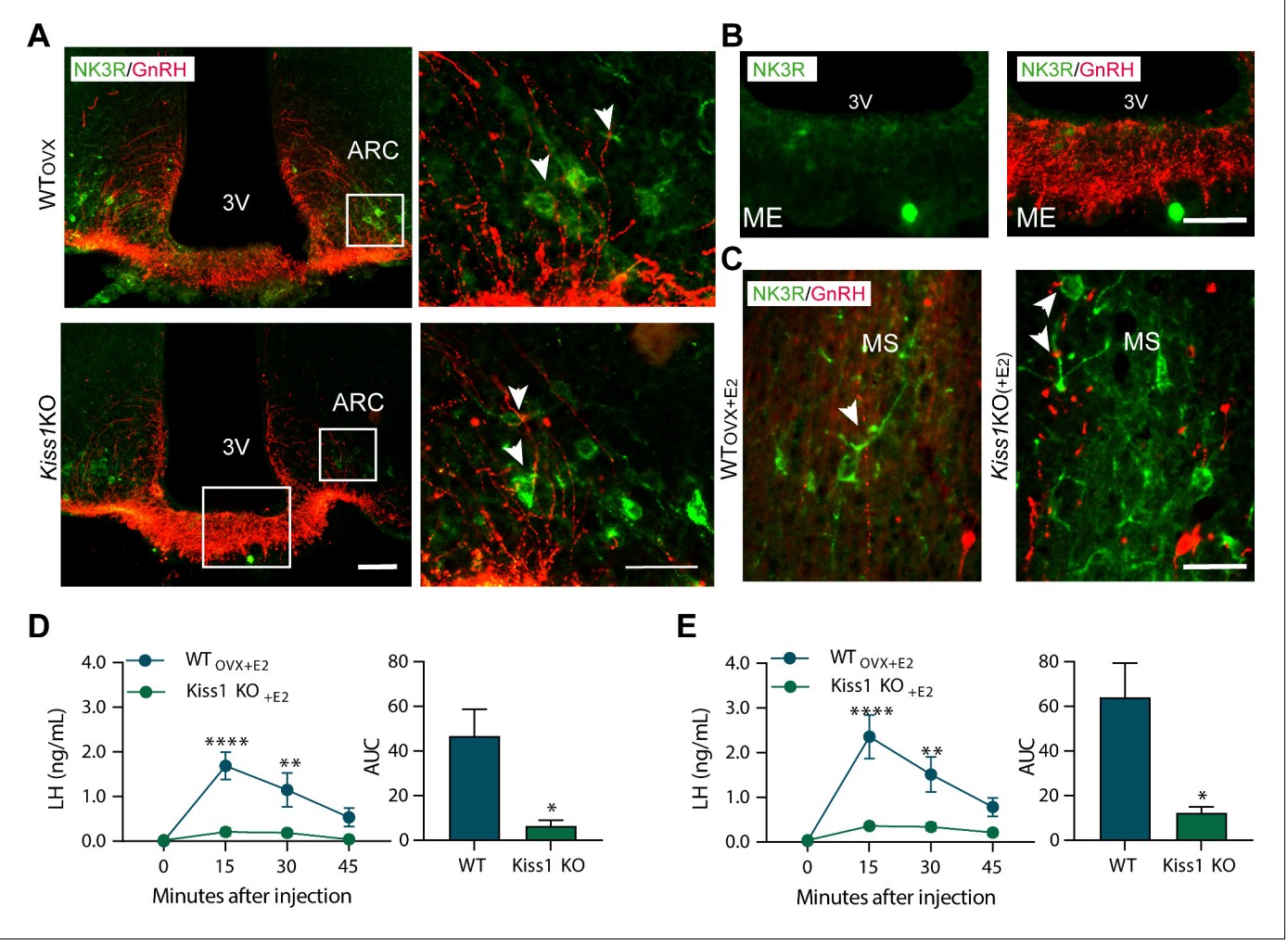

**Figure 3.** Senktide administration directly in to the arcuate nucleus or preoptic area (at the level of the medial septum) stimulates LH release in adult WT but not Kiss1$^{Cre/Cre}$ (i.e., *Kiss1* KO) female mice supplemented with estradiol. (A, B) Representative photomicrographs depicting dual label detection of NK3R (green) and GnRH (red) in the ARC (A) and ME (B) of ovariectomised WT (WTovx) and homozygous *Kiss1*$^{Cre/Cre}$ (i.e., *Kiss1* KO animals). (A) Panels on the right are enlarged images (scale bar: 20 μm) from boxed areas on the left (scale bar: 100 μm) showing infrequent close appositions (arrowheads) between NK3R and GnRH in the ARC. (B) Enlarged images of the boxed area from the *Kiss1* KO animal in (A) showing intense GnRH but lack of NK3R staining in the ME. (C) Dual-label detection of NK3R (green) and GnRH (red) in the POA and specifically the level of the MS of ovariectomized WT and homozygous *Kiss1*$^{Cre/Cre}$ (i.e., *Kiss1* KO) female mice supplemented with estradiol [WT$_{OVX+E2}$ and *Kiss1* KO$_{+E2}$, respectively; scale bar: 50 μm]. Arrowheads indicate infrequent sites of close apposition. (D, E) Mean ±SEM LH concentrations (ng/ml) and area under the curve (AUC) after an injection of senktide into the ARC (D) or POA (E) of ovariectomized WT (WT$_{OVX+E2}$; blue line-blue bar) and homozygous *Kiss1*$^{Cre/Cre}$ (i.e., *Kiss1* KO$_{+E2}$; green line-green bar) female mice supplemented with estradiol (n = 5/group). 3V: third ventricle, ARC: arcuate nucleus, ME: median eminence, POA: preoptic area, MS: medial septum. Repeated LH concentrations at multiple time-points and between treatments were compared using a 2-WAY ANOVA and a Fishers *posthoc* test when appropriate. Area under the curve was compared with 2-tailed student t tests. \*\*p < 0.0015, \*\*\*\*p < 0.0001.

DOI: https://doi.org/10.7554/eLife.40476.006

The following figure supplements are available for figure 3:

**Figure supplement 1.** NK3R antibody validation.
DOI: https://doi.org/10.7554/eLife.40476.007

**Figure supplement 2.** Representative photomicrographs depicting dual label detection of NK3R (green) and GnRH (red) in the ARC of WT$_{(OVX+E2)}$ and *Kiss1* KO$_{+E2}$ animals.
DOI: https://doi.org/10.7554/eLife.40476.008

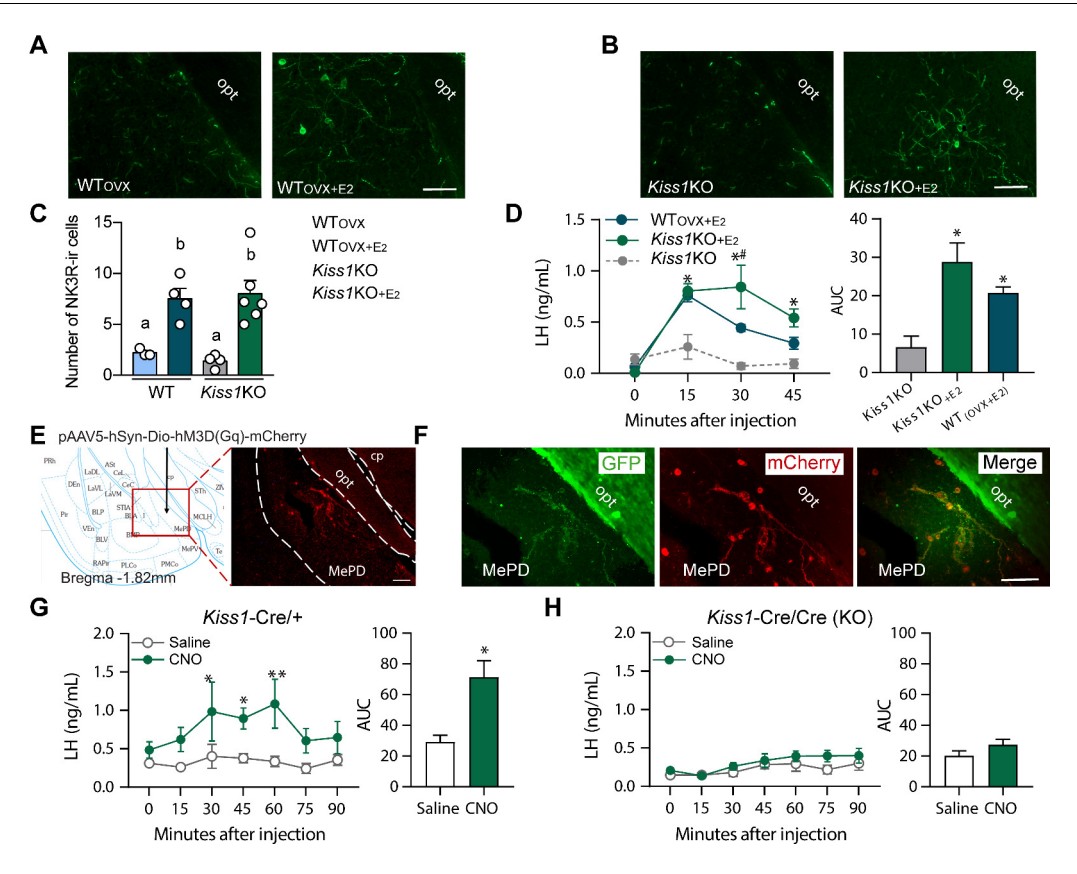

**Figure 4.** Senktide administration into the posterodorsal medial amygdala stimulates LH release in WT and *Kiss1* KO female mice supplemented with estradiol. (A, B) Representative photomicrographs depicting NK3R-immunoreactive cell bodies and fibers in the MePD of (A) ovariectomized WT (WT$_{OVX}$; left panel) or ovariectomized and estradiol-supplemented WT (WT$_{OVX+E2}$; right panel) and (B) homozygous *Kiss1*$^{Cre/Cre}$ (i.e., *Kiss1* KO; left panel) or *Kiss1* KO supplemented with estradiol (*Kiss1* KO$_{+E2}$; right panel) female mice. (C) Mean ±SEM number of NK3R-immunoreactive cells per 30 μm section in the MePD of WT$_{OVX}$, WT$_{OVX+E2}$, *Kiss1* KO and *Kiss1* KO$_{+E2}$ female mice (as quantified from 6 to 8 sections from 4 to 5 animals/group). (D) Mean ± SEM LH concentrations (ng/ml) and area under the curve (AUC) after an injection of senktide into the MePD of ovariectomized WT (WT$_{OVX+E2}$; blue line-blue bar) or homozygous *Kiss1*$^{Cre/Cre}$ (i.e., *Kiss1* KO$_{+E2}$; green line-green bar) supplemented with estradiol or hypogonadal (with low sex steroids) *Kiss1* KO female mice (n = 5/group). (E) Schematic representation of the site of injection of an AAV encoding a Cre-dependent hM3Dq: mCherry. The left panel is a schematic representation of the location of the MePD injection site and its anatomical relationship to the optic tract (−1.82 mm from bregma). The right panel is a higher magnification of the boxed area on the left showing MePD kisspeptin neurons tagged with mCherry (red fluorescence), which indicates hM3Dq receptor expressing kisspeptin neurones. (F) Representative photomicrograph of a coronal brain section stained for GFP (green), mCherry (red) and merged GFP and mCherry immunoreactivity in the MePD of a *Kiss1* KO female mouse >3 weeks after hM3Dq: mCherry injection (Scale bar 50 μm). In this animal model Cre and GFP have been targeted to the Kiss1 locus and are, therefore, expressed solely in Kiss1 neurons. Thus, colocalization of GFP and m-Cherry immunoreactivity represents Kiss1$^{ARC}$ cells that have been infected with the pAAV encoding a Cre-dependent hM3Dq DREADD. (G, H) Mean ± SEM LH responses and area under the curve (AUC) to an injection of saline (grey line-empty bar) or CNO (green line-green bar) of hM3Dq DREADD injected *Kiss1*$^{Cre/+}$ (G) and *Kiss1*$^{Cre/Cre}$ (KO; H) female mice (n = 5/group). opt: optic tract. Repeated LH concentrations at multiple time-points and between treatments were compared using a 2-WAY ANOVA and a Fishers *posthoc* test when appropriate. Area under the curve was compared with 2-tailed student t tests. *p < 0.025, **p < 0.0014.

DOI: https://doi.org/10.7554/eLife.40476.009

The following figure supplement is available for figure 4:

**Figure supplement 1.** Representative photomicrographs of a coronal section stained for NK3R (green), NKB (red) and merged NK3R and NKB immunoreactivity in the amygdala of (A) WT$_{(OVX+E2)}$ and NKB (green) of (B) WT$_{(OVX)}$ female mouse.

DOI: https://doi.org/10.7554/eLife.40476.010

Large NK3R immunoreactive neurons and fibers were identified in the substantia innominata, paraventricular nucleus, supraoptic nucleus, lateral hypothalamus, zona incerta, and perifornical regions. Interestingly, in the ARC, cells containing NK3R were evident only in WT$_{OVX}$ and hypogonadal *Kiss1* KO female mice (*Figure 3A* and *Figure 3—figure supplement 2*), whereas, in the MePD, NK3R cell

bodies appeared only when WT and *Kiss1* KO females were supplemented with $E_2$ (*Figure 4A, B and C*).

Next, we investigated the anatomical relationship of NK3R and GnRH expression with dual-label immunohistochemistry, which revealed minimal close appositions between the two proteins in the ARC and medial septum (MS) (*Figure 3A and C*). In these two areas, GnRH cells and fibers showed intermingling with NK3R containing cells and fibers, but no co-expression within GnRH cell bodies (>100 cells analyzed from a total of 16 mice; *Figure 3C*). Interestingly, we observed no fibers containing NK3R immunoreactivity in the internal or external zone of the median eminence (ME; *Figure 3A and B*).

To identify the brain area in which NK3R receptive neurons that mediate the kisspeptin-independent GnRH release reside, we stereotaxically administered senktide specifically into the ARC, the POA (at the level of the MS) or the MePD. These areas are prime candidates to play a role in LH stimulation because (a) they contain NK3R, the immunoreactivity of which is regulated by $E_2$, (b) there is an anatomical overlap of NK3R and GnRH protein, at least in the MS and ARC and (c) they contain GnRH and/or Kiss1 cell bodies and fibers and are known to play an important role in reproductive function (*Smith et al., 2006*; *Kim et al., 2011*). Senktide administration into the ARC or POA of $WT_{OVX+E_2}$ mice stimulated LH secretion within 15 min from drug infusion (*Figure 3D and E*; $p < 0.0001$ for both) compared to *Kiss1* KO animals. However, when senktide was administered into the MePD of $WT_{OVX+E_2}$ and *Kiss1* KO females supplemented with $E_2$ a robust increase in LH was observed within 15 min after senktide infusion that was similar in both genotypes (*Figure 4D*). Conversely, in the absence of $E_2$, *Kiss1* KO females did not show any alteration in LH release (*Figure 4D*), mimicking the LH responses we obtained after an ICV injection of senktide in these animals (*Figure 1B*).

## Chemogenetic activation of the MePD Kiss1 neuron stimulates LH release in WT but not Kiss1 KO female mice

Similar to the experiments described above, we delivered Cre-dependent AAV5-DIO-hM3Dq:mCherry to the MePD of $Kiss1^{Cre/+}$ or $Kiss1^{Cre/Cre}$ mice. HM3Dq:mCherry expression was present in the MePD (*Figure 4F*) and was limited to sections ranging from (from −1.6 mm to −2.0 mm from bregma; paxinos atlas). Two out of eight mice (one from each group) had primarily unilateral spread of the DREADD, but they did not differ significantly from their respective groups in LH concentrations, and were therefore, included in further analyses. Within the MePD, mCherry cell bodies were co-expressed in ~89% of GFP-immunoreactive cells (i.e. Kiss1 cells), and was not observed in non-GFP cells.

$Kiss1^{Cre/+}$ mice expressing hM3Dq:mCherry in the MePD and treated with CNO to activate the $Kiss1^{MePD}$ neurons, had an increase in LH within 30 min after the injection ($p = 0.0107$) compared to animals receiving saline treatment, which was sustained for another 30 min before returning to basal levels (*Figure 4G*). No alteration in LH was observed in $E_2$-supplemented *Kiss1* KO animals treated with either saline or CNO (*Figure 4H*).

## Discussion

Our results provide evidence that the MePD is a component of the gonadotropin axis. Specifically, we have identified two independent pathways within the MePD that can lead to the stimulation of GnRH/LH release. The first involves $Kiss1^{MePD}$ neurons, the activation of which, stimulates LH release into the peripheral circulation. Furthermore, this is achieved by the release of kisspeptin and not by any other signaling molecules produced within the $Kiss1^{MePD}$ neuron, since LH was increased only in animals with an intact kisspeptin signaling system. A second pathway, involving NKB/NK3R signaling, was also identified, when senktide (NK3R agonist) administration into the MePD induced LH release in *Kiss1* KO mice (*Padilla et al., 2018*). Thus, kisspeptin is not a required mediator between NK3R activation in the MePD and LH secretion. Interestingly, this pathway is female-specific and estrogen-dependent as responses were absent in males and hypogonadal females*Figure 1*.

From a mechanistic point of view, the most likely kisspeptin-independent pathway for LH stimulation by NKB would involve the direct regulation of GnRH release (*Krajewski et al., 2005*). Despite there being an anatomical overlap of GnRH and NK3R protein, specifically in the ARC and POA (at the level of the MS), we observed no instances of colocalization between NK3R and GnRH cell

bodies irrespective of the presence or absence of sex steroids. Furthermore, GnRH and NK3R cellular associations were minimal with scattered foci of close appositions in the ARC and the POA (at the level of the MS). This reveals certain anatomical differences to what has been previously demonstrated in the rat, where ~ 16% of GnRH cell bodies were found to contain NK3R protein (*Krajewski et al., 2005*). However, our results agree with reports of no NK3R expression in GnRH neurons of the ewe (*Amstalden et al., 2010*) suggesting the existence of species differences. Overall, our data suggest that the kisspeptin-independent action of NKB cannot be attributed to direct stimulation of NK3R located on GnRH neurons.

Kiss1$^{ARC}$ (KNDy) neurons make close appositions with GnRH cell bodies and terminals (*Lehman et al., 2010*) and may therefore stimulate GnRH neurons through intermediates other than kisspeptin. For example, it has been demonstrated with in vitro examination of coronal brain slices, that senktide induces GnRH release from the ME and this effect is, in part, present in *Kiss1* KO mice (*Gaskins et al., 2013*). We did not observe any NK3R immunoreactive fibers in the internal or external zone of the median eminence, indicating a potential lack of direct NKB (or senktide) regulation of the GnRH terminals in that area. Nonetheless, other signaling molecules such as glutamate (*Nestor et al., 2016*), galanin or γ-aminobutiric acid (*Skrapits et al., 2015*) can potentially stimulate LH secretion and must also be considered. However, activation of the Kiss1$^{ARC}$ (KNDy) neuron stimulated LH release only in mice with an intact kisspeptin signaling system and was completely absent in *Kiss1* KO mice. This provides evidence that kisspeptin, but no other signaling molecule produced by Kiss1$^{ARC}$ (KNDy) neurons can stimulate LH release in vivo. It must be noted that recent evidence has demonstrated that CNO is reverse-metabolized to clozapine in the bloodstream and exerts clozapine-like behavioral effects, thus challenging the assertion that CNO is pharmacologically inert (*Manvich et al., 2018*). However, this does not appear to be the case in the current Kiss1$^{Cre}$ mouse model as *Kiss1* KO mice were injected with an identical protocol as heterozygous (Kiss1$^{Cre/+}$) and yet showed no alteration in LH. Thus, we can conclude that non-specific clozapine effects are not an issue in this model.

The distribution of NK3R has been described in the human, rat, and ewe (*Mileusnic et al., 1999*; *Krajewski et al., 2005*; *Amstalden et al., 2010*) and here, we confirm a similar distribution in the mouse brain. Interestingly, in certain areas the immunoreactivity of NK3R-containing cell bodies was highly dependent on sex steroid levels. Specifically, estrogen downregulated NK3R expression in the ARC whereas the opposite was true for the MePD, with more NK3R containing cell bodies evident when animals were supplemented with E$_2$. High sensitivity of NK3R expression to E$_2$, has also been reported for the ARC with in situ hybridization studies (*Navarro et al., 2009*). Interestingly, this regulation of NK3R expression is reminiscent of the regulation of *Kiss1* by E$_2$ in these areas (*Kim et al., 2011*; *Smith et al., 2006*).

Based on the aforementioned anatomical observations we proceeded with senktide administration into the ARC, POA (at the level of the MS) and MePD of E$_2$-treated animals in order to locate the kisspeptin-independent, LH-stimulating population of NK3R-expressing neurons. Senktide administration into the ARC or POA (at the level of the MS) significantly stimulated LH secretion in WT females compared to *Kiss1* KO animals. Therefore, our data suggest that LH release, as a result of NKB/NK3R signaling in the POA or ARC, is predominantly achieved via initial kisspeptin release, involving Kiss1$^{AVPV/PeN}$ and/or Kiss1$^{ARC}$ neuron activation. Indeed, 10% of Kiss1$^{AVPV/PeN}$ and virtually all Kiss1$^{ARC}$ neurons contain NK3R (*Navarro et al., 2015*). Moreover, both populations are interconnected, Kiss1$^{ARC}$ cells project to Kiss1$^{AVPV/PeN}$ neurons and GnRH cell bodies and terminals (*Yip et al., 2015*), which could also account for the increase in LH after administration of senktide into the POA. However, a slight increase in LH occurred in the absence of kisspeptin signaling (*Kiss1* KO mice). In the ARC and MS of the POA, we observed few instances were NK3R containing fibers were in close apposition to GnRH cell bodies and/or processes indicating a minimal, but potentially present, presynaptic action of NKB onto GnRH. It is possible that NKB (or senktide) signaling onto presynaptic NK3R, results in the enhanced secretion of other stimulatory neuropeptides, which in turn stimulate GnRH secretion and could account for the slight increase in LH observed in *Kiss1* KO mice, as documented in the rat striatum, in which tachykinins (including NKB) presynaptically stimulate the release of dopamine (*Glowinski et al., 1993*).

Interestingly, senktide administration into the MePD of females supplemented with E$_2$ produced a similar robust increase in LH in animals with or without the presence of the *Kiss1* gene. Conversely, hypogonadal *Kiss1* KO females did not show any alteration in LH release, indicating that this

kisspeptin-independent NKB/NK3R signaling mechanism in the MePD becomes activated only when $E_2$ is present. This notion is further supported by our finding that the number of NK3R cells increases with $E_2$, and this upregulation is specific to the MePD. Furthermore, these NK3R expressing cells do not co-localize with NKB, but are surrounded by a plethora of NKB fibers (*Figure 4—figure supplement 1*) the source of which remains to be determined. Likely candidates include NKB-expressing cells residing within the ARC or the neighboring NKB population in the central amygdala (CeA) (*Figure 4—figure supplement 1*) to date, implicated only in the modulation of fear memories (*Andero et al., 2016*).

To investigate the mechanism further, we determined whether MePD NK3R neuron activation is mediated directly or indirectly by Kiss1^MePD neurons through the release of signaling molecules other than kisspeptin (since senktide stimulates LH release in *Kiss1* KO mice). Chemogenetic activation of the Kiss1^MePD neuron provided evidence that the activation of *Kiss1* neurons in the MePD of the female mouse can stimulate LH but only through the release of kisspeptin. This, clearly demonstrates the influence of Kiss1^MePD signalling on the gonadotropic axis. Furthermore, GnRH/LH stimulation via the Kiss1^MePD neuron is most likely not part of the kisspeptin-independent NKB/NK3R signaling pathway but of a second LH stimulating mechanism originating from the MePD. To date, functional studies in rodents have shown that an injection of kisspeptin specifically in to the amygdala results in increased LH secretion, while blocking endogenous amygdala kisspeptin signalling with a kisspeptin antagonist decreases both LH secretion and LH pulsatility (*Comninos et al., 2016*) indicating that the kisspeptin released may act locally, within the MePD, as part of the LH stimulating pathway. Thus, we cannot rule out the possibility that NK3R neurons in the MePD also contain Kiss1r and are activated by both NKB and Kiss1.

The functional relevance of either the MePD kisspeptin-dependent or the kisspeptin-independent neuronal population that can induce LH release is unknown. A reasonable hypothesis is that Kiss1^MePD and NK3R^MePD neurons are part of the neuronal network linking pheromonal/social cues and gonadotropin release (*Yang et al., 2018*). Indeed, estrogen receptors are expressed in the MePD (*Lymer et al., 2018*) and the brain region is a central hub for processing sensory inputs such as olfactory signals and integrating these into behavioral (*Rajendren and Moss, 1993*; *Adekunbi et al., 2018*) and neuroendocrine outputs (*Pineda et al., 2017*). Specifically, it is compelling to hypothesize that the kisspeptin-independent, NK3R-dependent pathway is employed for the generation and/or enhancement of the LH surge and/or female sexual behavior, for example lordosis, given that this mechanism was absent in male mice, and is exclusively activated in the presence of estrogen, similar to what is observed in the female AVPV/PeN (*Smith et al., 2006*). In accordance, recent evidence demonstrated the enhancement of the LH surge in rats exposed to male-soiled bedding, which was accompanied by an increased Fos expression in Kiss1^AVPV/PeN neurons as well as various limbic structures, including the MePD (*Watanabe et al., 2017*; *Hellier et al., 2018*).

In summary, we have shown that the MePD is a previously unknown component of the gonadotropic axis. Initially, we observed that senktide administration into the lateral ventricle stimulates LH release into the peripheral circulation of female mice lacking kisspeptin (*Kiss1* KO), but only when they are supplemented with $E_2$. Upon further investigation, we identified two mechanisms that can lead to GnRH/LH secretion in the female, involving Kiss1^MePD or NK3R-expressing neurons located in the MePD (*Figure 5*). Collectively, these data demonstrate that the gonadotropic axis is subject to regulation by signalling originating outside the hypothalamus and specifically the MePD, involved in the regulation of social behaviors including sexual behavior, anxiety, and olfaction.

## Materials and methods

### Animals

A *Kiss1^Cre:GFP* knock-in mouse (version 2) was generated from C57Bl/6 blastocysts and verified at the University of Washington (*Padilla et al., 2018*). The homozygous version of this mouse, *Kiss1^Cre:Cre* is a *Kiss1* KO as characterized elsewhere (*Padilla et al., 2018*). *Kiss1* gene was also confirmed undetectable from POA and MBH tissue (*Figure 1—figure supplement 1*). Animals were group housed according to sex and bred under constant conditions of temperature (22–24°C) and light [12 hr light (06:00)/dark (18:00) cycle], fed with standard mouse chow (Teklad F6 Rodent Diet 8664) and were given *ad libitum* access to tap water. For all studies, C57Bl/6 WT or *Kiss1^Cre/+* (heterozygous state)

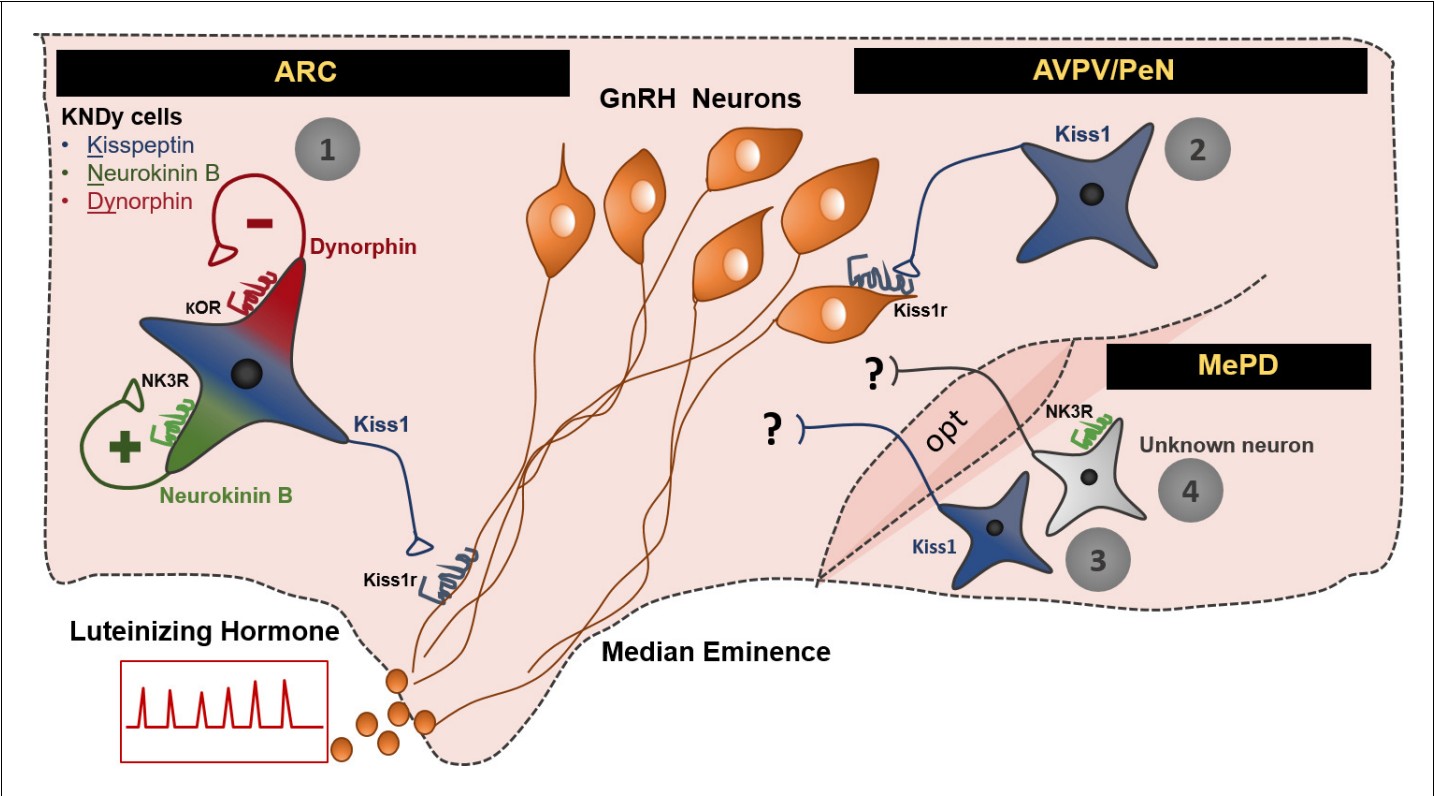

**Figure 5.** Schematic representation of the neuronal pathways known to stimulate GnRH/LH release in the female mouse. The first pathway involves Kiss1 neurons located in the arcuate nucleus (ARC), which also express neurokinin B and dynorphin (thus, often referred to as KNDy neurons). These cells have been mainly implicated in the regulation of GnRH/LH pulsatile secretion in males and females. The second GnRH/LH stimulating mechanism, originates from Kiss1 cells located in the anteroventral periventricular/periventricular preoptic nucleus (AVPV/PeN) and is mainly implicated in the generation of the GnRH/LH surge, which leads to ovulation, and is specific to females. The third and fourth stimulating mechanisms are described in this study and originate from the posterodorsal medial amygdala (MePD). Thus, the third mechanism known to stimulate GnRH/LH release involves Kiss1 neurons and the fourth NK3R expressing neurons of unknown phenotype. The latter are activated only in the presence of estradiol in a mechanism that is female specific. The precise neuronal pathway linking these MePD cells with the stimulation of GnRH/LH release, as well as the physiological relevance of this stimulation, remains to be elucidated. ARC: Arcuate Nucleus, AVPV/PeN: anteroventral periventricular/periventricular preoptic nucleus, MePD: posterodorsal medial amygdala, NK3R: neurokinin receptor 3, κOR: κ opioid receptor, Kiss1r: kiss1 receptor, opt: optic tract.
DOI: https://doi.org/10.7554/eLife.40476.011

males or females between age 8 and 20 weeks were used and studied in parallel to *Kiss1^{Cre/Cre}* (*Kiss one* knock-out state) littermates. In order to test the specificity of the NK3R antibody, NK3RKO mice were used as described below.

## Experiment 1: Effect of central (ICV) administration of senktide on LH release in male and female WT and Kiss1 KO mice with or without the presence of sex steroids.

In this experiment we aimed to assess whether central activation of NK3R signaling with senktide (an NK3R specific agonist), can stimulate LH release in *Kiss1 KO* mice (i.e., in a kisspeptin-independent manner) in the presence or absence of sex steroids. Adult WT male and female mice were GND and studied in parallel to hypogonadal (with low sex steroid levels) *Kiss1 KO* littermates (n = 7–10/ group). ICV injections (see below) of senktide (Tocris Bioscience, Cat. No. 1068; 600 pmol diluted in 5 µl 0.9% NaCl) were performed and blood samples were collected before (basal) and 25 min after ICV injection for LH measurements as has been previously described (*Navarro et al., 2015*). Next, animals were implanted with sex steroids (n = 5–10/group) and the ICV experiment was repeated a week later. The dose of senktide used, and the time of blood collection were selected based on our previous studies (*Navarro et al., 2015*).

## Experiment 2: Effect of ARC KNDy neuron chemogenetic activation on LH release in WT and Kiss1 KO female mice in the presence of estradiol

In order to determine whether the release of other components, besides kisspeptin, within the Kiss1$^{ARC}$ (KNDy) neuron can stimulate LH release, we used a chemogenetic approach to specifically activate Kiss1$^{ARC}$ neurons of $Kiss1^{Cre/+}$ or $Kiss1^{Cre/Cre}$ mice treated with E$_2$ (n = 5–8/group). Females received bilateral stereotaxic injections (see below) of an adeno-associated virus (pAAV) encoding a Cre-driven Gq-coupled DREADD (pAAV5/hSyn-DIO-hm3Dq:mCherry; Addgene, Cat. No.44361-AAV5; titer 3 × 10$^{12}$ genome copies per ml; 1 µl per hemisphere). Following infection, mice were given 3 weeks for recovery and maximum expression of the AAV vector. On the day 1 of the experiment animals were administered an ip bolus injection of vehicle saline (0.9% NaCl; day 1) and then hM3D receptors were activated by ip injection of its agonist, clozapine N-oxide (CNO; 10 mg/kg dissolved in saline; day 2). Blood samples were collected just before saline or CNO treatment (0) and then every 15 min for 90 min. The dose of CNO was chosen based on previous behavioral studies using hM3Dq manipulations (*Ben-Shaanan et al., 2016*). At the end of the experiment, all mice were treated with an icv injection of senktide, as a control, and to confirm that animals were appropriately treated and primed.

## Experiment 3: Effect of senktide administration in to the ARC, POA or MePD on LH release in female WT and Kiss1 KO mice with the presence of estrogen.

In this set of experiments, we aimed to locate the brain area which senktide is acting to stimulate LH release. To this end, we first conducted neuroanatomical studies to confirm NK3R protein expression in the mouse hypothalamus, as well as to investigate the potential anatomical interplay between NK3R and GnRH neurons, as the most plausible kisspeptin-independent mechanism. Thus, WT$_{OVX}$, WT$_{OVX+E2}$, $Kiss1 KO$ and $Kiss1 KO_{+E2}$ (n = 5/group) were perfused following standard protocols and the brains were collected for immunohistochemical (IHC) analyses, as described in detail below. Based on IHC results, a stereotaxic injection approach was used to specifically activate NK3R in the POA (at the level of the MS), or ARC or MePD and monitor LH responses in the peripheral circulation of anesthetized WT$_{OVX+E2}$ and $Kiss1 KO_{+E2}$ females. Unilateral injections were performed as described below on WT$_{OVX+E2}$, and $Kiss1 KO_{+E2}$ (n = 5/group) which received 600 pmol of senktide diluted in 1 µl saline (0.9% NaCl) in the POA or ARC or MePD. An additional control group was added to the MePD injected cohort which consisted of $Kiss1 KO$ mice without E$_2$ treatment (n = 5), which according to results from experiment one should not lead to an increase in LH after senktide administration. Blood samples were collected before (0) and then every 15 min for 45 min after senktide administration. The first blood sample (15 min post administration) was taken with the needle still in place. Lastly, animals were decapitated and the brains collected, frozen on dry ice and stored at −80°C for injection site confirmation (see below).

## Experiment 4: Effect of MePD Kiss1 neuron chemogenetic activation on LH release in WT and Kiss1 KO female mice in the presence of estradiol

To determine whether the release of other components, besides kisspeptin, within the Kiss1$^{MePD}$ neuron can stimulate LH release, we used a similar approach as previously described to specifically activate Kiss1$^{MePD}$ neurons of $Kiss1^{Cre/+}$ or $Kiss1^{Cre/Cre}$ mice treated with E$_2$ (n = 5/group). E$_2$ is known to upregulate $Kiss1$ expression in the MePD and $Cre:GFP$ expression follows an identical pattern in this mouse model (*Padilla et al., 2018*). Thus, hypogonadal $Kiss1^{Cre/Cre}$ mice injected with the Cre-dependnet hM3Dq:mCherry in the MePD were also treated with an E$_2$ capsule prior to surgery. The remainder of the experimental protocol was similar to what has been described in experiment 2.

## Gonadectomy and sex steroid replacement

The effects of sex steroids or lack thereof on LH secretion was established via bilateral GND; (*Ström et al., 2012*; *Idris, 2012*) of adult male and female WT mice (WT$_{ORX}$ and WT$_{OVX}$) with circulating sex steroid levels being restored between genotypes (WT$_{ORX+T}$, WT$_{OVX+E2}$, $Kiss1 KO_{+T}$ and $Kiss1 KO_{+E2}$) via subcutaneous implantation of capsules (1.5 cm long, 0.078 in inner diameter, 0.125 in outer diameter; Dow Corning) containing 50 µg/ml 17$\beta$-estradiol (Sigma-Aldrich), in sesame oil or

testosterone in powder form (1 cm filled area). Neuroendocrine experiments were consistently conducted 7 d after gonadectomy or sex steroid supplementation (*García-Galiano et al., 2012*).

## Kiss1 KO GnRH priming

To exclude the possibility that absence in gonadotropin responses to the various stimuli in hypogonadal *Kiss1 KO* mice may result from inadequate pituitary responsiveness to GnRH, which has been previously described in animals with a defective *Kiss1* signaling system (*Roa et al., 2008*) *Kiss1 KO* mice were subjected to a protocol of GnRH priming during 2 days prior testing, as has been previously described (*García-Galiano et al., 2012*). In this protocol, each mouse received five successive ip boluses of a low dose of GnRH (0.15 µg/each), with the following schedule: at 10:00 hr, 17:00 hr, and 23:50 hr on the first day; at 0800 and 1600 on the second day with neuroendocrine tests being conducted on the third day (*García-Galiano et al., 2012*). WT mice injected with saline vehicle, following the same protocol, served as controls.

## Intracerebroventricular (ICV) Injections

ICV injections were performed following preciously published procedures (*Navarro et al., 2015*). Briefly, 2–3 days before the experiment, mice were anesthetized with isoflurane and a small hole was bored in the skull 1 mm lateral and 0.5 mm posterior to bregma with a Hamilton syringe attached to a 27-gauge needle fitted with polyethylene tubing, leaving 2.0 mm of the needle tip exposed. Once the initial hole was made, all subsequent injections were made at the same site. On the day of ICV injection experiments, mice were anesthetized with isoflurane for a total of 5–10 min, during which time 5 µl of solution were slowly and continuously injected into the lateral ventricle. The needle remained inserted for approximately 30 s after the injection to minimize backflow up the needle track. Mice typically recovered from the anesthesia within 3 min after the injection.

## Stereotaxic injections

Mice were deeply anaesthetized with isoflurane and placed into a stereotaxic apparatus (Kopf Instruments, Model 940). After exposing the skull via incision, a small hole was drilled for injection at the appropriate AP and ML coordinates. A syringe (Hamilton, 5 µL, Model 175 RN SYR, 32 ga, Cat. No.80016) was lowered into the brain at the appropriate DV coordinates. Coordinates relative to bregma were as follows: anteroposterior (AP) −1.6 mm, mediolateral (ML) ± mm and dorsoventral (DV) −5.85 mm for the ARC, AP +0.6 mm, ML ±0.25 mm, DV −5.15 mm for the POA and AP −1.9 mm, ML ±2.0 mm, DV −4.9 mm for the MePD. Injection sites were chosen based on the Paxinos Brain Atlas, and confirmed with India Ink (Fisher Scientific, Cat. No. NC9903975) trial injections. Each infusion was slowly delivered over 2 min (500 nl/min), the needle was left in place for an additional 5 min (for AAV injections) and 15 min (for senktide administrations) and then slowly withdrawn to minimize backflow. Animals received 0.3 mg/kg buprunex (subcutaneous) during surgery and 24 hr later for analgesia and were allowed a 3 week recovery before onset of experiments.

## Blood samples and LH measurements

In all cases blood samples for LH measurements were obtained after a single excision of the tip of the tail. The tip was cleaned with saline and then massaged prior to taking a 4 µl blood sample from the tail tip with a pipette. Whole blood was immediately diluted in 116 µl of 0.05% PBST [phosphate buffer saline (Boston Bio Products, Cat. No. BM220) containing Tween-20 (Sigma, Cat. No. P2287), vortexed, and frozen on dry ice. Samples were stored at −80°C until analyzed with LH ELISA (*Steyn et al., 2013*).

## Immunohistochemistry

### Tissue preparation

Animals were terminally anesthetized with a ketamine/xylazine in saline (0.9% NaCl) cocktail and transcardially perfused with 0.1 M phosphate-buffer (0.1M PB) followed by 4% paraformaldehyde diluted in 0.1M PB (PFA; Electron Microscopy Sciences). Brains were removed, stored in the same fixative for 4 hr and then transferred into sucrose solution [Thermo Fisher Scientific; 20% sucrose in 0.1 M PB containing 0.01% sodium azide (Sigma-Aldrich)] at 4°C. After sucrose infiltration tissue was cut into 30 µm coronal sections using a freezing stage microtome (Fisher HM440E). The tissue

sections were separated into two groups of three parallel series (90 μm apart). The first group consisted of sections extending from the medial septal nucleus to the caudal part of the retrochiasmatic area (+1.0 mm to −1.0 mm relative to bregma; containing GnRH cell bodies and the RVP3V population of $Kiss1$ cells) and the second encompassing the ARC nucleus (from −1.0 till −2.8 mm relative to bregma; containing ARC and amygdala $Kiss1$ populations of cells). Sections were stored at −20°C in cryoprotectant [30% sucrose in 0.1 m PB containing 30% ethylene glycol (Thermo Fisher Scientific) and 0.01% sodium azide] until further processing.

## General procedures

For all staining procedures detailed below, free-floating sections at room temperature and under gentle agitation were thoroughly washed in PBS, pH 7.4, between all incubations, and treated with 10% $H_2O_2$ (10 min; in PBS) and PBS$^+$ [1 hr; PBS containing 0.1% bovine serum albumin (Thermo Fisher Scientific) and 0.4% Triton X-100 (Sigma-Aldrich)]. Brain sections were mounted onto Superfrost plus glass slides (Fisher Laboratories), air dried and cover-slipped with Vectashield HardSet Mounting Medium (Vector Laboratories). For immunodetection of the different proteins, sections of all animals were processed simultaneously. Secondary antibodies were tested for non-specific binding by primary antibody omission. mCherry immunostaining was compared and validated with the endogenous mCherry expressed by the viral construct in a separate series of sections. The anti-GnRH and anti-NKB antibodies produced staining patterns identical to those from several independent GnRH and NKB antibodies (*Merlo et al., 2007*; *Krajewski et al., 2005*) and consistent with those by in situ hybridization (*Duarte et al., 2006*). The specificity of the NK3R antibody was tested by staining brain tissue collected from NK3R KO mice (*True et al., 2015*) alongside experimental tissue. No NK3R staining was observed in NK3RKO animals (*Figure 3—figure supplement 1*).

### mCherry and GFP

Brains from $Kiss1^{Cre/+}$ and $Kiss1^{Cre/Cre}$ mice (n = 5/group) injected with AAV5-hSyn-DIO-hM3D(Gq)-mCherry were assessed for mCherry reporter expression to confirm bilateral infection of ARC or MePD GFP expressing $Kiss1$ neurons with the AVV carrying Cre-dependent hM3D receptor. One series of free-floating sections containing ARC and amygdala from each animal was incubated overnight in blocking solution containing rat anti-mCherry primary antiserum, Alexa Fluor 594 conjugate (1:500; Thermo Fisher, Cat. No. M11240). The next morning sections were extensively washed in PBS and then incubated overnight in rabbit anti-GFP tag antibody (1:5,000; Thermo Fisher, Cat. No. A-6455) followed by goat-anti-rabbit DyLight 488 secondary antibody (1:200; Thermo Scientific, Cat. No. 35552).

### NK3R and GnRH

Brains from WT and Kiss1 KO mice were processed for NK3R and GnRH in order to compare the protein distribution of NK3R in the two genotypes and investigate the potential colocalization with GnRH in the hypothalamus. Furthermore, we processed tissue sections from WT$_{(OVX)}$, WT$_{(OVX+E2)}$, $Kiss1$ $KO$ and $Kiss1$ $KO_{+E2}$ groups to determine whether $E_2$ regulates protein expression of NK3R in the a) preoptic area/medial septum, where most of the GnRH cell bodies are located b) ARC where KNDy cells reside, and c) MePD where senktide induced LH release. Hence, a series of every third section, extending from the level of the optic chiasma to the mammillary nucleus, was processed for NK3R and GnRH using a modified protocol previously described (*Goodman et al., 2007*). Tissue sections were incubated sequentially with: 1) rabbit anti-NK3R (1:30,000; Novus Biologicals, Cat. No. NB300-102) for 17 hr, 2) biotinylated goat anti rabbit IgG (1:500; Vector Laboratories, Cat. No. BA1000), for 1 hr, 3) avidin and biotinylated horseradish peroxidase complex (Avidin-Biotin Complex; 1:500; Vector Laboratories, Cat. No. PK-6100) for 1 hr, 4) biotinylated tyramine (1:250; PerkinElmer, Cat. No. NEL700A001KT), containing 0.003% $H_2O_2$ for 10 min, and 5) DyLight 488 conjugated streptavidin (1:200; Thermo Fisher, Cat. No. 35552) for 30 min. Next, sections were incubated with rabbit anti-GnRH (1:1,000; Abcam; Cat. No. ab5617) for 17 hr. The next morning, sections were washed and incubated with goat anti-rabbit Alexa 555 (1:200; Thermo Fisher, Cat. No. A-21428) for 30 min. The specificity of the NK3R antibody we used in our anatomical studies (see below) was tested by staining brain tissue collected from transgenic mice with mutations in the NKB receptor and previously described and validated [NK3R-/- mice(*True et al., 2015*).

### NK3R and NKB

An additional series of sections containing MePD from WT$_{(OVX+E2)}$ mice was processed for staining with NK3R as described above. Sections were further incubated overnight with rabbit-anti Neurokinin B (1:1000; Novus Biologicals, Cat. No. NB300-201) and goat anti-rabbit Alexa 555 (1:200) for 30 min.

## Microscopy and image analysis

### Validation of senktide injection site

The locations of POA, ARC and MePD injection sites were investigated in sections cut at 20 μm thickness using a cryostat (Fisher, HM505E). Every other section was collected around the injection site, mounted on microscope slides air-dried and cover slipped with Vectashield HardSet Mounting Medium (Vector Laboratories, Burlingame, CA). Only animals with accurate and restricted injection sites were included in the analysis.

### Validation of chemogenetic activation of ARC KNDy or MePD Kiss1 neurons

Sections from animals injected with AAV vectors encoding hM3Dq:mCherry in the ARC or MePD were examined, and the location of mCherry expression was confirmed in GFP positive neurons. In both cases, quantification of GFP and GFP/mCherry positive neurons in all areas was carried out in a subpopulation of animals (n = 4/group) with images taken at x20 magnification from two representative sections per animal.

### NK3R/GnRH anatomical relationship

The anatomical relationship between NK3R/GnRH (throughout the hypothalamus), mCherry/GnRH (in the ARC) and NK3R/NKB (in the MePD) was examined in sections 90 μm apart, from each mouse. In addition, comparisons of NK3R cell numbers between WT$_{OVX}$, WT$_{OVX+E2}$, *Kiss1 KO* and *Kiss1 KO*$_{+E2}$ (n = 5/group) were performed in six to eight sections at 20X magnification containing ARC and MePD, to determine the effect of E$_2$ on protein expression in these areas. Counts were averaged per mouse, per brain area. A digital camera (CoolSnap EZ, PhotometricsTM, Canada) attached to a microscope (Nikon Eclipse 90i), with the appropriate excitation for DyLight 488 (green flurescence) and Alexa 555 (red fluorescence) and NIS-Elements Viewer AR 310 software was used to examine tissue sections and superimpose two images and determine putative colocalization or interactions. Montages of images and adjustments of brightness and contrast levels were made in Adobe Photoshop CS5.

## Quantitative real-time RT PCR

We aimed to confirm the lack of *Kiss1* expression in the POA and MBH of *Kiss1 KO* mice. WT (OVX; n = 3) and *Kiss1* KO (n = 5) female mice were killed, brains were exposed and the POA and MBH was extracted and immediately frozen in dry ice and stored at −80 C. Total RNA from both areas was isolated using TRIzol reagent (Invitrogen) followed by chloroform/isopropanol extraction. RNA was quantified using NanoDrop 2000 spectrophotometer (Thermo Scientific) and one microgram of RNA was reverse transcribed using Superscript III cDNA synthesis kit (Invitrogen). Quantitative real-time PCR assays were performed in triplicates of each sample on an ABI Prism 7000 sequence detection system, and analyzed using ABI Prism 7000 SDS software (Applied Biosystems). The cycling conditions were as follows: 2 min incubation at 50°C, 10 min incubation at 95°C (hot start), 40 amplification cycles (95°C for 15 s, 60°C for 1 min, and 45 s at 75°C, with fluorescence detection at the end of cycles 3–40), followed by melting curve of the amplified products obtained by ramped increase of the temperature from 55°C to 95°C to confirm the presence of single amplification product per reaction. The data were normalized using L19 primers as an internal control. *Kiss1* expression was detected using primers: F- CTCTGTGTCGCCACCTATGC R – TTCCCAGGCATTAACGAGTTC. Values were normalized with housekeeping gene *Rpl19*.

## Immunohistochemistry for mCherry and GnRH *mCherry and GnRH*

Brains from *Kiss1*[Cre/+] and *Kiss1*[Cre/Cre] mice (n = 5/group) injected with AAV5-hSyn-DIO-hM3D(Gq)-mCherry were assessed for mCherry reporter expression and GnRH to confirm anatomical integrity of *Kiss1* neuron and GnRH fiber interaction in the area. A potential explanation for the lack of LH

responses in DREADD-injected *Kiss1 KO* mice could be developmental alterations in the projections from KNDy neurons to GnRH neurons (*Clarkson and Herbison, 2006*). Furthermore, that could explain the lack of LH stimulation after chemogenetic activation of Kiss1 neurons in *Kiss1 KO* mice. The anatomical relationship between the two proteins appeared to be similar in both genotypes (*Kiss1*$^{Cre/+}$ and *Kiss1 KO*), suggesting that kisspeptin is not needed developmentally for the formation of fibers in Kiss1 neurons. (*Figure 2—figure supplement 1*). One series of free-floating sections containing ARC and amygdala from each animal was incubated overnight in blocking solution containing rat anti-mCherry primary antiserum, Alexa Fluor 594 conjugate (1:500; Thermo Fisher, Cat. No. M11240). The next morning sections were extensively washed in PBS and then incubated overnight in rabbit anti-GnRH (1:1,000; Abcam, Cat. No. ab5617) followed by goat-anti-rabbit DyLight 488 secondary antibody (1:200; Thermo Scientific, Cat. No. 35552).

## Statistics

All data are presented as mean ± SEM. Single point comparisons (basal LH *versus* after ICV injection), were made using 2-tailed paired *t* tests. Repeated LH concentrations at multiple time-points and between treatments were compared using a 2-WAY ANOVA and a Fishers *posthoc* test when appropriate. Area under the curve was compared with 2-tailed student *t* tests. A *P* value less than 0.05 was considered significant. All analyses were performed with GraphPad Prism Software, Inc (San Diego, CA).

## Study approval

All animal care and experimental procedures were approved by the National Institute of Health, and Brigham and Women's Hospital Institutional Animal Care and Use Committee, protocol #05165. The Brigham and Women's Hospital is a registered research facility with the U.S. Department of Agriculture (#14–19), is accredited by the American Association for the Accreditation of Laboratory Animal Care and meets the National Institutes of Health standards as set forth in the Guide for the Care and Use of Laboratory Animals (DHHS Publication No. (NIH) 85–23 Revised 1985).

## Acknowledgments

We thank Dr. Stephanie Seminara for kindly providing NK3RKO mice for the validation of the NK3R antibody. This work was supported by NIH R00 HD071970, R01 HD090151 and Women's Brain Initiative to VMN.

## Additional information

### Competing interests

Richard D Palmiter: Reviewing Editor, eLife. The other authors declare that no competing interests exist.

### Funding

| Funder | Grant reference number | Author |
| --- | --- | --- |
| National Institutes of Health | R01 HD090151 | Victor M Navarro |
| National Institutes of Health | R00 HD071970 | Victor M Navarro |
| Brigham and Women's Hospital | Women's Brain Initiative | Victor M Navarro |

The funders had no role in study design, data collection and interpretation, or the decision to submit the work for publication.

### Author contributions

Chrysanthi Fergani, Conceptualization, Data curation, Software, Formal analysis, Validation, Investigation, Visualization, Methodology, Writing—original draft, Writing—review and editing, Conceived

and designed the research, Conducted experiments, Contributed to data analysis; Silvia Leon, Formal analysis, Investigation, Methodology, Conducted experiments, Contributed to data analysis; Stephanie L Padilla, Resources, Validation, Methodology, Writing—original draft, Generated and provided the Kiss1Cre mice; Anne MJ Verstegen, Resources, Investigation, Methodology, Conducted experiments; Richard D Palmiter, Conceptualization, Resources, Validation, Writing—original draft, Generated and provided the Kiss1Cre mice; Victor M Navarro, Conceptualization, Resources, Data curation, Software, Formal analysis, Supervision, Funding acquisition, Validation, Investigation, Visualization, Methodology, Writing—original draft, Project administration, Writing—review and editing, Conceived and designed the research, Contributed to data analysis

### Author ORCIDs
Chrysanthi Fergani  https://orcid.org/0000-0001-7028-4158
Richard D Palmiter  https://orcid.org/0000-0001-6587-0582
Victor M Navarro  http://orcid.org/0000-0002-5799-219X

### Decision letter and Author response
Decision letter https://doi.org/10.7554/eLife.40476.014
Author response https://doi.org/10.7554/eLife.40476.015

## Additional files

### Supplementary files
• Transparent reporting form
DOI: https://doi.org/10.7554/eLife.40476.012

### Data availability
All data generated or analysed during this study are included in the manuscript and supporting files.

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
