## [Decision Letter]

[**Editorial note:** This article has been through an editorial process in which the authors decide how to respond to the issues raised during peer review. The Reviewing Editor's assessment is that all the issues have been addressed.]

Thank you for submitting your article "NKB Signaling in the Medial Amygdala Stimulates Gonadotropin Release in a Kisspeptin-Independent Manner in Female Mice" for consideration by *eLife*. Your article has been reviewed by three peer reviewers, one of whom is a member of our Board of Reviewing Editors, and the evaluation has been overseen by Catherine Dulac as the Senior Editor. The following individual involved in review of your submission has agreed to reveal his identity: Sebastien G Bouret (Reviewer #3). Two other reviewers remain anonymous.

The Reviewing Editor has highlighted the concerns that require revision and/or responses, and we have included the separate reviews below for your consideration. If you have any questions, please do not hesitate to contact us.

Summary:

Fergani and colleagues describe results investigating the kisspeptin and neurokinin b systems in regulating LH secretion. They report that NKB increases LH secretion in a kisspeptin-independent manner. Moreover, they report a previously unrecognized role of kisspeptin neurons in the medial amygdala in regulating LH secretion in female mice. Collectively, the studies are well described, use multi-disciplanary approaches, and provide novel data. A few issues need to be addressed.

Major concerns:

As outlined below, the major concern is the relative lack of neuroanatomical information. This needs to be addressed to allow the reader to assess the data fully.

Separate reviews (please respond to each point):

*Reviewer #1:*

My biggest criticism of the manuscript is the way the data are presented which is somewhat confusing. This is critical for a journal with a wide readership. For example, a major finding appears to be identifying a role of the MeA kisspeptin neurons in regulating LH secretion. This appears to be almost an add on and is not highlighted in the Abstract. It is unclear to this reviewer how this fits with the kisspeptin-independent results with the NKB agonist.

Similarly, a summary figure of the proposed model would help clarify the new findings.

*Reviewer #2:*

NKB is generally thought to function with kisspeptin in the regulation of gonadotropin secretion. In this manuscript, the authors describe a series of anatomical and pharmacological experiments in mice that further characterize the role of the NKB/NK3R signaling in the regulation of LH secretion. Specifically, the authors examined whether the presence of estradiol or testosterone influence the effect of the NK3R agonist, senktide, on LH secretion in a previously characterized kisspeptin loss of function mouse model (Kiss1KO). In this work, they found that when estradiol is present, senktide increases LH secretion in Kiss1KO mice (i.e. in a kisspeptin independent manner). This effect was only observed in female mice indicating the effect is sex-specific. Additionally, the authors provide evidence suggesting that this kisspeptin-independent effect of senktide on LH secretion occurs in the medial amygdala. Overall, the findings from this work contribute to what is known about NKB signaling and LH regulation. The NKB system has always remained less well-characterized than the kisspeptin system, and up to this point it was generally believed that NKB only influenced GnRH and LH secretion by acting through kisspeptin. The data reported in this paper suggest that NKB can act independently of kisspeptin in the medial amygdala, which provides a strong rationale for new lines of research. While the results in the manuscript are generally worth reporting, there are both major and minor concerns that need to be addressed.

Of particular concern is the inadequate documentation of experimental findings. The images provided in nearly all figures are poorly formatted and of low quality. This undermines confidence in the quantitative measures and the specificity of experimental interventions. All of the images ostensibly demonstrating cellular associations are not convincing. Equally troubling are images meant to illustrate successful DREADD expression and signaling in the medial amygdala (which part?), which do not provide strong support for the authors statements in the manuscript (Figure 4 and Figure 4—figure supplement 1). Furthermore, key reagents should be clearly described and their validation detailed. For example, the effectiveness of the viral transductions used to express DREADDs should be examined beyond mere confirmation of expression of the mCherry reporter, perhaps by confirming neuronal activation using an independent signal (e.g. Fos). The figure legends are remarkably uninformative and need extensive revision. A glaring example is that neither the text in the Results section, nor the legend for Figure 2, clearly describes what the GFP labeling represents.

Minor issues:

1) If there is not a reference limit, the authors should cite the original manuscripts in the Introduction instead of Fergani and Navarro 2016.

2) In the Results section beginning in the fourth paragraph, the authors should briefly explain why they decided to investigate the distribution of NK3R. The way it is written now, this section seems to come out of nowhere and doesn't really fit with the rest of the experiments. I know some of this rationale was given in the Materials and methods section, but since it will be at the back of the article, it's possible some readers will never see it.

3) In the legend for Figure 1, change the formatting of (C, D).

4) In the legend for Figure 2, indicate that the Kiss1^cre/cre^ mice in figure D are treated with estradiol as is stated in the text.

5) The manuscript should be carefully revised to eliminate errors in syntax and awkward sentence structure.

Additional data files and statistical comments:

Adequate information on data analysis, however the authors should display individual data points on bar graphs, not just SEM, which can be misleading. Rigour of experimental approach addressed in general comments.

*Reviewer #3:*

This is a well-written manuscript on a timely and interesting topic. While a great deal of research in the reproductive field has focused on Kisspeptin/NKB neurons in the hypothalamus, little is known about the extra-hypothalamic action of those neurons and their distinct contribution in reproductive function. This paper uses the full modern genetic armamentarium to elegantly identify novel neuronal pathways by which NKB controls gonadotropin secretion.

There is only one minor issue that should be addressed: the authors use CNO to chemoactivate kisspeptin neurons. While CNO is undoubtely a powerful activator of muscarinic DREADDs, its metabolic conversion of CNO to clozapine is a potential concern since clozapine can have its own biological/physiological effects. The authors should at least discuss this potential concern.

---

## [Author Response]

As outlined below, the major concern is the relative lack of neuroanatomical information. This needs to be addressed to allow the reader to assess the data fully.

In order to address this concern we have improved the quality of the images, re-taken new ones in the cases where the quality was not sufficient, and added new anatomical landmarks to help the reader locate the area of study. In addition, we have added a model summarizing our findings in new Figure 5.

Separate reviews (please respond to each point):

Reviewer #1:

My biggest criticism of the manuscript is the way the data are presented which is somewhat confusing. This is critical for a journal with a wide readership. For example, a major finding appears to be identifying a role of the MeA kisspeptin neurons in regulating LH secretion. This appears to be almost an add on and is not highlighted in the Abstract. It is unclear to this reviewer how this fits with the kisspeptin-independent results with the NKB agonist.

Thank you for pointing this out. We agree that kisspeptin derived stimulation of LH originating from the amygdala is an important finding. As such, it should be highlighted in the Abstract. Furthermore, we have rephrased accordingly and described that the reasons why we decided to test the activation of the Kiss1^MePD^ neuron (to determine whether NK3R activation may result from the release of other signaling molecules besides kisspeptin). Please see Abstract, Introduction second and fourth paragraphs and Discussion paragraph seven.

Similarly, a summary figure of the proposed model would help clarify the new findings.

Thank you for the suggestion. We have added a figure summarizing our findings. Please see new Figure 5.

Reviewer #2:

[…] Of particular concern is the inadequate documentation of experimental findings. The images provided in nearly all figures are poorly formatted and of low quality. This undermines confidence in the quantitative measures and the specificity of experimental interventions. All of the images ostensibly demonstrating cellular associations are not convincing.

We appreciate this reviewer’s concern and to address it, we have extensively reformatted the images by improving their resolution, adding landmarks for better understanding of the area that was studied and added new images when needed to further clarify the neuroanatomical region of study.

The images assessing cellular association are intended to show *lack* of detectable colocalization between GnRH and NK3R. Therefore, this is the reason why no clear association between them is observed in the photographs and the reason why we discuss that GnRH is *NOT* directly activated by senktide. We have clarified this point in the revised text and figure legends. Please see paragraph three in subsection “Senktide administration into the MePD, but not the ARC or POA, stimulates LH release in female WT and Kiss1 KO mice, in the presence of estrogen.”, Discussion paragraphs two and five and new figure 3 legend).

Equally troubling are images meant to illustrate successful DREADD expression and signaling in the medial amygdala (which part?), which do not provide strong support for the authors statements in the manuscript (Figure 4 and Figure 4—figure supplement 1).

We have revised the images to delineate the specific part of the MeA that was studied (MePD), and added a lower magnification image for better neuroanatomical characterization of the area injected with the DREADD. Please see new Figure 4.

Furthermore, key reagents should be clearly described and their validation detailed. For example, the effectiveness of the viral transductions used to express DREADDs should be examined beyond mere confirmation of expression of the mCherry reporter, perhaps by confirming neuronal activation using an independent signal (e.g. Fos).

All of the reagents used in the study as well as their validation are described in the material and methods section. The expression of mCherry in Kiss1 neurons is cre dependent by the nature of its DIO (AAV5-nSyn-DIO-hM3D(Gq)-mCherry) construct. We believe that this, together with the specific action of CNO in DREADD injected animals, is sufficient to prove the effectiveness of the viral transduction.

The figure legends are remarkably uninformative and need extensive revision. A glaring example is that neither the text in the Results section, nor the legend for Figure 2, clearly describes what the GFP labeling represents.

Thank you for the comment. We have extensively revised the figure legends and added a description of what GFP represents both in the figure legends and in the text. Please see paragraph one of subsection “Chemogenetic activation of the ARC KNDy neuron stimulates LH release in control but not Kiss1 KO female mice” and new Figure 2 and Figure 4 legends.

Minor issues:1) If there is not a reference limit, the authors should cite the original manuscripts in the Introduction instead of Fergani and Navarro 2016.

This has been corrected in the revised manuscript.

2) In the Results section beginning in the fourth paragraph, the authors should briefly explain why they decided to investigate the distribution of NK3R. The way it is written now, this section seems to come out of nowhere and doesn't really fit with the rest of the experiments. I know some of this rationale was given in the Materials and methods section, but since it will be at the back of the article, it's possible some readers will never see it.

Thank you for bringing in up this issue. A clear rationale for the study of NK3R has been added to the Results section, subsection “Senktide administration into the MePD, but not the ARC or POA, stimulates LH release in female WT and Kiss1 KO mice, in the presence of estrogen”.

3) In the legend for Figure 1, change the formatting of (C, D).

Thank you, corrected.

4) In the legend for Figure 2, indicate that the Kiss1^cre/cre^ mice in figure D are treated with estradiol as is stated in the text.

Thank you. Figure legend has been revised accordingly.

5) The manuscript should be carefully revised to eliminate errors in syntax and awkward sentence structure.

The manuscript has been revised to address these issues.

Additional data files and statistical comments:Adequate information on data analysis, however the authors should display individual data points on bar graphs, not just SEM, which can be misleading. Rigour of experimental approach addressed in general comments.

We have made the necessary changes to all of our bar graphs.

Reviewer #3:

[…] There is only one minor issue that should be addressed: the authors use CNO to chemoactivate kisspeptin neurons. While CNO is undoubtely a powerful activator of muscarinic DREADDs, its metabolic conversion of CNO to clozapine is a potential concern since clozapine can have its own biological/physiological effects. The authors should at least discuss this potential concern.

We thank this reviewer for their positive comments on the manuscript. We agree with his concern and this has been further addressed in the manuscript. The fact that KO mice, injected with the same protocol of CNO treatment did not respond with the LH increase seen in het mice provides evidence that there was no toxic or unspecific effect of CNO in this model. Please see Discussion paragraph three.